# Effect of Replacing Feldspar by Philippine Black Cinder on the Development of Low-Porosity Red Stoneware

Fel Jane A. Echavez [1], Liberty R. Lumasag [1], Beverly L. Bato [1], Alyssa May Rabadon Simplicio [1], Jade P. Cahigao [1], Elly U. Aligno, Jr. [1], Roben Victor M. Dispo [1], Sherlyn Keh D. Dionio [1], Christian Julle C. Saladaga [1], Raymond V. Rivera Virtudazo [1,2] and Ivyleen C. Bernardo-Arugay [1,2,*]

[1] Research Center for Advanced Ceramics, College of Engineering and Technology, Mindanao State University—Iligan Institute of Technology, Iligan City 9200, Philippines

[2] Ceramic Engineering Program, Department of Materials and Resources Engineering and Technology, College of Engineering and Technology, Mindanao State University—Iligan Institute of Technology, Iligan City 9200, Philippines

[*] Correspondence: ivyleen.arugay@g.msuiit.edu.ph

**Abstract:** Stoneware is a ceramic material with low porosity and high mechanical properties, such as the modulus of rupture. It is essentially made of clay, feldspar and quartz and is sintered to create a mixture of glass and crystalline phases. With the projected growth rate of the global ceramics market size and the country's development plan for 2023–2028, it is imperative that alternative raw materials for the manufacture of ceramic products be sourced so that the importation of these materials, such as feldspar, be minimized, if not eliminated. Cinder in the Philippines is mainly used as a filling material in pavements and residential areas. In this study, this resource is utilized as partial and full replacement of feldspar in a typical ternary diagram for stoneware production. Bars were formed from different formulations by the slip casting method and were sintered at 1200 °C. Physical and mechanical properties of the bars, such as shrinkage, loss on ignition, water absorption, apparent porosity and modulus of rupture were determined. Thermo-physical analyses were also carried out on the raw materials and on formulated powders. Meeting the requirements of the various quality standards for ceramics, the partial replacement of feldspar with black cinder (LF, LFBQ and LFBH) is feasible for wall and roof applications while full replacement of feldspar with black cinder (LB) is suitable for wider use as wall, floor, vitrified, industrial and roof tiles.

**Keywords:** cinder; flux; stoneware; low-porosity ceramics; thermo-physical properties of ceramics; mechanical properties of ceramics; volcanic tuff; red clays; anorthite; andesine; quartz; feldspar; smectites

## 1. Introduction

In 2020, the novel coronavirus (COVID-19) was declared as a global pandemic by the World Health Organization. This pandemic greatly affected the world economy with a rise in global poverty for the first time in 20 years [1–3]. Philippines had its deepest economic contraction at −16.9% in the second quarter of 2020, with underemployment hitting a high of 21% in the middle of 2021. To accelerate the economic and social recovery from the pandemic-induced losses, the country's development plan for 2023 to 2028 is geared on the underlying theme of economic and social transformation towards an inclusive and resilient society, taking off from the lessons learned especially during pandemic. One of the agendas of this transformation plan is to promote trade and investments in order to improve the competitiveness of domestic industries, increase demand for Philippine products and generate more jobs [4–6]. Among the promising industries in the country is the ceramic tile industry, whose organization, the Ceramic Tile Manufacturers' Association (CTMA), is a member of the international group Ceramic Industry Club of ASEAN (CICA) [7]. With the continuing expansion of the country's infrastructure program [8,9], abundant opportunities remain for the ceramic manufacturing industries.

The global ceramics market size is projected to reach USD 379.27 billion by 2030, from a value of USD 244.69 billion in 2021, at a compound annual growth rate (CAGR) of 4.48% within the forecast period of 2022 to 2030. The rising construction activity globally as well as home renovations are driving the market for ceramics [10]. With the rising demand for ceramic products, there would be greater need for the raw materials for the manufacture of tiles, bricks, sanitary wares, art wares, pipes and other advanced ceramic applications in electronics, telecommunications, manufacturing, transportation, medicine, defense and space exploration. The most common raw material in the ceramics industry is feldspar for its use as a fluxing agent. Its low melting point which forms a vitreous phase during firing contributes to the sintering of a given mixture [11–13]. Studies have been made on possible replacement of feldspar in the production of ceramic tiles with materials such as metallurgical slag [14], fly ash [15], waste glass [16], CRT panel glass [17], volcanic ash [18], coffee husk ash [19] and biomass bottom ash [20]. This is in anticipation of the possible exhaustion of feldspar as well as creating a wider source of raw materials for the increasing demand of ceramic products such as stoneware.

Stoneware bodies are widely used in sanitaryware, dinnerware, art wares, pottery, and floor, wall and roof tiles. The main components of a stoneware body are clay, feldspar and quartz, and it is sintered at 1190–1230 °C to create a mixture of glass and crystalline phases [21,22]. The stoneware body has low water absorption (<3%) [23,24], high mechanical strength ($\geq$30 MPa) [23,25] and resistance to deep abrasion of less than 175 mm$^3$ [26,27].

With the projected growth rate of the global ceramics market size and the country's development plan, it is imperative that alternative raw materials for the manufacture of ceramic products be sourced. This is to minimize the importation of these materials, such as feldspar, into the country. It is with this study that possible replacement of feldspar with black cinder is presented. The black cinder of Salvador, Lanao del Norte, located in the southern part of the Philippines, is used as filling a material in roads and residential areas within the region.

Cinder gravels are pyroclastic materials that come in different sizes and varying colors such as black, red, gray or brown. They are weak materials and have high porosity, which make them unsuitable for use as base course materials, especially for heavily trafficked roads. These properties however can be improved with methods, such as cement stabilization, to make them suitable for base course construction materials [28–30].

In this study, the black cinder of Salvador, Lanao del Norte will be explored as an alternative raw material to commercial feldspar in the manufacture of ceramic products. Another resource of the region is the Linamon red clay, which will be utilized as the source of the clay content of stoneware according to a typical ternary formulation of stoneware [31–33]. To the best of our knowledge, this is the first work to report the potential of black cinder from the Philippines as partial or full replacement of feldspar in the production of stoneware. The feasibility of its use will reduce the dependence of ceramic manufacturing industries on imported feldspar and optimize the utilization of the resource in the region. This will greatly support the development plan of the country in promoting Philippine products through utilizing the available resources of the land, generating more employment with the strengthening of ceramic manufacturing industries and enhancing inter-sectoral linkages. All of these will answer the call for sustainable development of the United Nations in ending poverty, promoting decent work and economic growth as well as building sustainable cities and communities [34]. It is the aim of the study to determine the chemical composition and mineralogical properties of Lanao Salvador black cinder (LSBC), the morphology and thermal behavior of pure and formulated slips, and the characterization of the physical and mechanical properties from different formulations. Furthermore, the water absorption and modulus of rupture of formulated tiles will be evaluated using various standards [35–39].

## 2. Materials and Methods

### 2.1. Raw Materials

The formulations investigated in this study were prepared from the following raw materials: red clay from Linamon (LLRC) and black cinder from Salvador (LSBC), all in the region of Lanao del Norte, Philippines. The as-received raw materials were dried to remove the moisture and then crushed by a UA V-Belt Drive pulverizer, (BICO Braun International, Burbank, CA, USA). Pulverized samples were then dry screened using a 100-Tyler mesh sieve to remove coarse particles with a size greater than 150 microns.

### 2.2. Raw Material Characterization

Chemical analyses of the raw materials were obtained by X-ray fluorescence (Olympus Innov-X Pro X-ray Fluorescence Spectrometer, Olympus Innov, Woburn, MA, USA) with an analytical range of Mg and above. The mineralogical phase characterization was carried out by X-ray diffraction techniques (InXitu BTX II XRD Analyzer, InXitu, Mountain View, CA, USA) with the following analysis conditions: K$\alpha$ radiation of copper anode ($\lambda$ = 3.54 Å), voltage of 40 kV, current of 30 mA and a measuring angle of 5–55° (2$\theta$). Results of X-ray powder diffraction of each raw material were primarily based on "figure of merit" values from available references [40–42]. The thermal stability of each raw material was evaluated by Shimadzu DTG-60H thermogravimetric analyzer (Shimadzu, Kyoto, Japan) to investigate its physical and chemical changes during heating. A sample weighing approximately 50–112 mg was heated to 1000 °C with a heating rate of 10 °C/min in an air atmosphere. The morphology and chemical composition of the raw materials with a particle size of less than 150 microns were determined by SEM-EDS on a JEOL JSM-6510 Series Scanning Electron Microscope (JEOL Ltd., Tokyo, Japan). The samples were sputter coated with platinum for 40 s at a current of 40 mA prior to testing.

### 2.3. Sample Preparation and Sintering

The preparation of the slip started with slaking the 150-micron passing powders of Lanao Linamon red clay (LLRC). The slip was blunged at a speed of 740 rpm by a drill press mixer Powerhouse PH-5132 (Powerhouse, Memphis, USA) to make a homogenous LLRC–water slip. After homogenizing the slip, it was left to age overnight or for at least 15 hours. The aged slip was wet screened using a 100-mesh sieve to ensure no coarse particles greater than 150 microns were present in the slip and to maintain or ensure homogeneity of the slip. Powders of LSBC and commercial feldspar (CF) were added to the homogenous slip of LLRC based on the formulated binary components LF which contains LLRC and CF only.

To satisfy the oxide requirement of the LF formulation, the Lanao Salvador black cinder (LSBC) gradually replaced the CF to evaluate its effect on the stoneware. The replacement of the commercial feldspar with the LSBC resulted in three formulations with their corresponding empirical formula shown in Table 1. The mixture was thoroughly blended using the drill press mixer at a speed of 740 rpm for 10 min to attain homogeneity. The fluidity of the slip was adjusted by the addition of sodium silicate and the density was maintained at 1.8 $\pm$ 0.01 g/cm$^3$. These formulated slips were left to age overnight for further homogenization and stability.

**Table 1.** Formulation of Porcelain Stoneware Tiles.

| Formulation | Empirical Formula |
|---|---|
| LF | 0.93 MgO• 1.55 CaO• Al$_2$O$_3$• 5.61 SiO$_2$• 0.15 Fe$_2$O$_3$ |
| LFBQ | 0.99 MgO• 1.29 CaO• Al$_2$O$_3$• 5.32 SiO$_2$• 0.18 Fe$_2$O$_3$ |
| LFBH | 1.05 MgO• 1.06 CaO• Al$_2$O$_3$• 5.05 SiO$_2$• 0.20 Fe$_2$O$_3$ |
| LB | 1.16 MgO• 0.64 CaO• Al$_2$O$_3$• 4.58 SiO$_2$• 0.25 Fe$_2$O$_3$ |

The aged slips were further mixed using a drill press mixer for 10 min prior to casting and then poured into the test bar mold. The mold was continuously filled with slip until a

slight separation of the cast from the mold was noticed, signaling that the cast is ready for demolding. Each of the test bars was marked with a 100 mm line using a vernier caliper as reference in the determination of its linear shrinkage. The test bars were dried at 110 °C until constant weight was attained. Sintering was then followed at 1200 °C in an electric muffle furnace (SH Scientific, SH-FU-36MHSH Scientific Co., Ltd., Sejong, Korea) with $45 \pm 5$ min soaking at maximum temperature. The furnace was then turned off and the samples were allowed to cool naturally.

### 2.4. Samples Characterization after Sintering

The thermal stability of the different formulations was also studied. A sample powder from each formulation weighing approximately 50 mg was subjected to thermal behavior using a thermogravimetric analyzer Shimadzu DTG-60H to investigate its physical and chemical changes during heating. The 150-micron sample was placed in an alumina crucible and heated to 1000 °C with a heating rate of 10 °C/min under ambient environment. The sintered test bars were meanwhile tested for their physical and mechanical properties, such as total linear shrinkage (TLS), loss on ignition (LOI), water absorption (WA), apparent porosity (AP), and modulus of rupture (MOR). The modulus of rupture was determined using a Universal testing machine (UTM) Shimadzu AGS-X ( Shimadzu, Duisburg, Germany) with a maximum load capacity of 5000 N. Testing was performed using three-point bending fixtures with the machine. Constant force was applied to the sample until its failure. The equations used to achieve the aforementioned properties can be found in published papers [12,35,43,44]. The results of water absorption and the modulus of rupture of the bars were evaluated using the four available standards for the various stoneware products, specifically for floor, wall, vitrified, industrial and roof tiles such as ISO standard 13006 [37], Philippine National Standard (PNS) [38], Indian Council of Ceramic Tiles and Sanitaryware (ICCTAS) [39], and Brazilian Association of Technical Standards (ABNT) [35,36].

## 3. Results and Discussions

### 3.1. Raw Material Characterization

3.1.1. X-ray Fluorescence (Chemical Composition)

The chemical composition of each raw material is given in Table 2, which primarily consists of $SiO_2$, $Al_2O_3$, CaO and MgO. Raw materials contain minor to trace amounts of NiO, $Cr_2O_3$, MnO and $TiO_2$. Both LLRC and LSBC have relatively high amounts of iron oxide ($Fe_2O_3$). The amount of alumina in the LLRC is higher than that of usual red clays for tile [45], terracotta [46,47] and traditional ceramic production [48,49] employing the slip casting method. LLRC is also found to have a significantly different chemical composition to nickel laterite mine waste (NMW) [45] which was found to be a good source of clay content for tile production. On the other hand, the amount of iron oxide in the LSBC is higher than that in CF and has lower flux content compared to CF. Furthermore, LSBC is classified as siliceous with pozzolanic properties as per ASTM C618 [50] where it has a content ($SiO_2$+ $Al_2O_3$ + $Fe_2O_3$) greater than 70% and less than 10% CaO. Results also show a flux/silica ratio of 0.4330 for CF and 0.3297 for LSBC, a flux/alumina ratio of 2.0857 for CF and 1.0706 for LSBC, and the alumina/silica ratio of 0.2076 for CF and 0.3080 for LSBC.

**Table 2.** Chemical Composition of Raw Materials.

| Mass % | $SiO_2$ | $Al_2O_3$ | $Fe_2O_3$ | $K_2O$ | MgO | CaO | NiO | $Cr_2O_3$ | MnO | $TiO_2$ |
|---|---|---|---|---|---|---|---|---|---|---|
| LLRC | 42.63 | 34.39 | 14.37 | — | 6.1 | — | 0.03 | 0.1 | 0.2 | 2.19 |
| CF | 60.65 | 12.59 | 0.31 | — | 5.16 | 21.12 | — | — | 0.02 | 0.15 |
| LSBC | 55.68 | 17.15 | 6.42 | — | 9.75 | 8.61 | 0.02 | 0.04 | 0.1 | 2.23 |

Note: "—" means below detection limit.

### 3.1.2. X-ray Diffraction (XRD) Analysis

The XRD patterns of each raw material are shown in Figure 1. From these results, it was detected that LLRC had various minerals, such as nontronite, montmorillonite, lizardite, forsterite, halloysite, andesine, anorthite, hematite, goethite and quartz, as shown in Figure 1a. On the other hand, CF contains minerals such as albite, calcite, forsterite and quartz. Meanwhile, LSBC had various minerals identified, as shown in Figure 1c, minerals that were predominantly detected include anorthite, andesine, montmorillonite, illite, nontronite, lizardite, augite and quartz. Quartz is the common mineral present in the three raw materials. Minerals such as montmorillonite, lizardite, anorthite, andesine and quartz present in LLRC and LSBC are few of the minerals that were also found in silts from Philippine nickel laterite mine waste [45,51,52], which is also from the southern Philippines. The summary of the minerals with their corresponding formula is presented in Table 3.

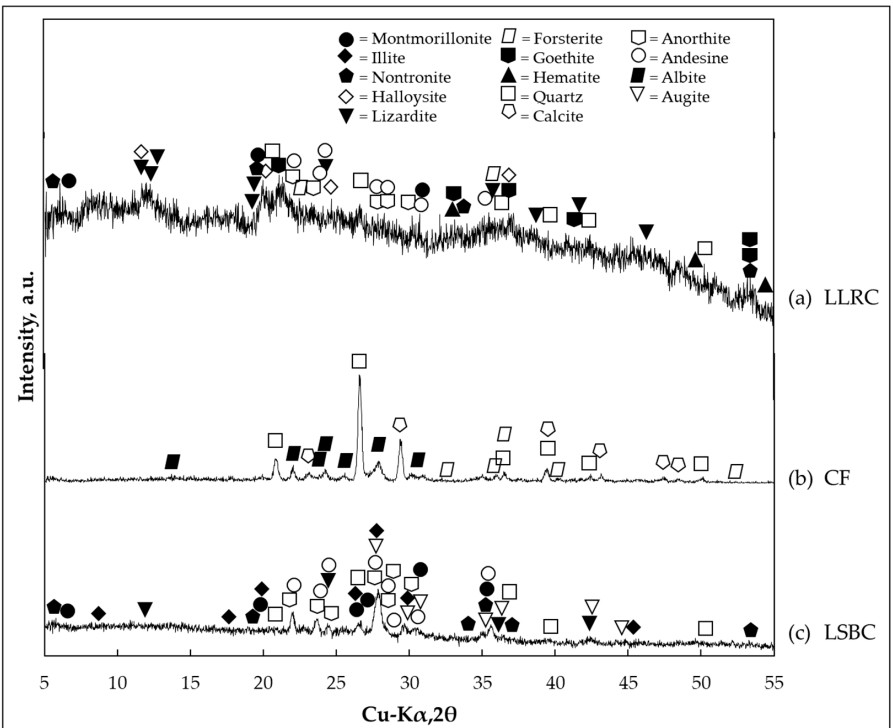

**Figure 1.** XRD patterns of raw materials: (**a**) LLRC; (**b**) CF; and (**c**) LSBC.

**Table 3.** List of detected minerals for LLRC, CF and LSBC.

| Mineral | Chemical Formula | LLRC | CF | LSBC |
|---|---|---|---|---|
| Montmorillonite | $(Na, Ca)_{0.3}(Al, Mg)_2Si_4O_{10}(OH)_2 \bullet nH_2O$ [40] | * | — | * |
| Illite | $(K,H_3O)(Al,Mg,Fe)_2(Si,Al)_4O_{10}[(OH)_2,(H_2O)]$ [40] | — | — | * |
| Nontronite | $Ca_{0.5}(Si_7Al_{0.8}Fe_{0.2})(Fe_{3.5}Al_{0.4}Mg_{0.1})O_{20}(OH)_4$ [40] | * | — | * |
| Halloysite | $Al_2Si_2O_5(OH)_4$ | * | — | — |
| Lizardite | $Mg_3Si_2O_5(OH)_4$ | * | — | * |
| Forsterite | $Mg_2SiO_4$ | * | * | — |
| Goethite | $Fe^{3+}O(OH)$ | * | — | — |
| Hematite | $Fe_2O_3$ | * | — | — |
| Quartz | $SiO_2$ | * | * | * |
| Calcite | $CaCO_3$ | * | * | — |
| Anorthite | $CaAl_2Si_2O_8$ | * | — | * |
| Andesine | $(Na,Ca)(Si,Al)_4O_8$ | * | — | * |
| Albite | $NaAlSi_3O_8$ | — | * | — |
| Augite | $(Ca,Mg,Fe)_2(Si,Al)_2O_6$ | — | — | * |

Note: "*" means minerals present, "—" means not detected.

### 3.1.3. Microscopy Analysis

The SEM micrographs of the raw materials are presented in Figure 2. The structure of LLRC is composed of irregular shaped loose flakes with rough edges, which are common for smectites [53–55], as shown in Figure 2a. On the other hand, the LSBC image shows glass shards with quasi-conchoidal fractures for some granular-shaped particles, as seen in Figure 2b. Platy structures with quasi card-of-house structures were also found, in addition to blocky structures, as shown in the yellow insets in Figure 2b. These platy or sheet structures could be attributed to the phyllosilicates present in LSBC.

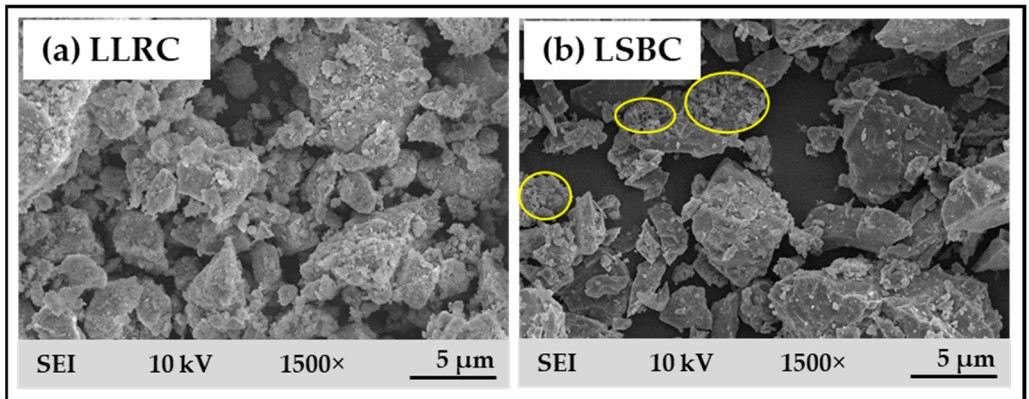

**Figure 2.** SEM micrographs of raw materials at 1500× magnification: (**a**) Lanao Linamon Red Clay (LLRC); (**b**) Lanao Salvador Black Cinder (LSBC) with yellow insets showing the platy structures.

### 3.1.4. Energy Dispersive X-ray (EDX) Analysis

The EDX (energy dispersive X-ray) spectra of the raw materials were shown in Figure 3, as well as the elemental analysis. The four major elements present in the spectrum are oxygen (O), silicon (Si), iron (Fe) and aluminum (Al), with minor to trace amounts of calcium (Ca) and sodium (Na), as shown in Figure 3c,d. It can be observed that LSBC contains elements of magnesium (Mg) and potassium (K), which are not present in LLRC.

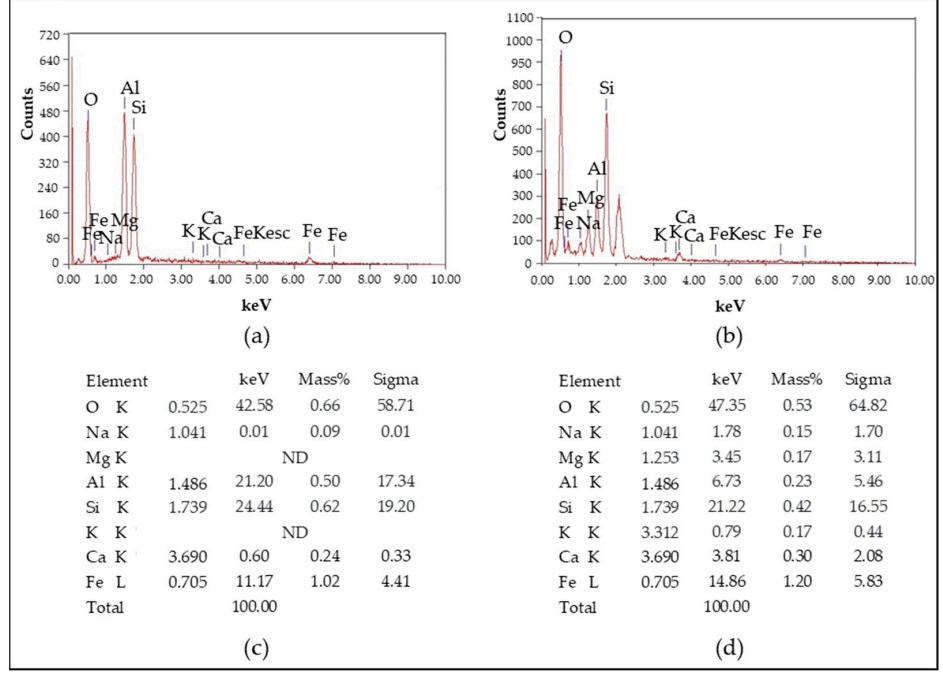

**Figure 3.** EDX analysis of raw materials: (**a**) spectra of LLRC; (**b**) spectra of LSBC; (**c**) Elemental Composition of LLRC; (**d**) Elemental Composition of LSBC.

### 3.2. Thermal Analysis of Raw Materials and Formulations

Results on the TGA-DTA of the three raw materials are shown in Figure 4. The total weight losses from heating LLRC, CF and LSBC are 20.49%, 9.44% and 0.88%, respectively, as indicated in Figure 4a. The DTA curve measured from LLRC has three endothermic reactions and two exothermic reactions, as shown in Figure 4b. The first endothermic peak is observed at 97.62 °C and is related to the elimination of free water [12,43,56,57]. The second endothermic reaction is observed at 287.62 °C which is due to the pre-dehydration of clay minerals [43]. The third endothermic reaction is at 525.37 °C, which is attributed to the dehydroxylation of minerals [12,43,56]. The first exothermic reaction is observed at 313.76 °C and is related to the burning off of carbonaceous matter [12]. An exothermic peak is also seen at 939.72 °C which is due to decomposition of montmorillonite and the formation of defect such as aluminum-silicon spinel ($Si_3Al_4O_{12}$) structures (structural reorganization), and some amorphous phases [57–60]. On the other hand, CF exhibited an endothermic reaction at 786.39 °C, as shown in Figure 4b, which is related to the decomposition of calcite [61,62], corroborating the presence of calcite in the XRD results. Meanwhile, no distinct phase transition is observed on the DTA curve of LSBC up to 700 °C (Figure 4d); however at the temperature range 700–900 °C, two exothermic peaks are observed at 737 °C and 840 °C, which are due to the oxidation phenomenon, and might also be attributed to the allotropic transformation of iron oxide [63,64] in the LSBC. The curve might also indicate overlapping of reactions that created a trough with a wide temperature range.

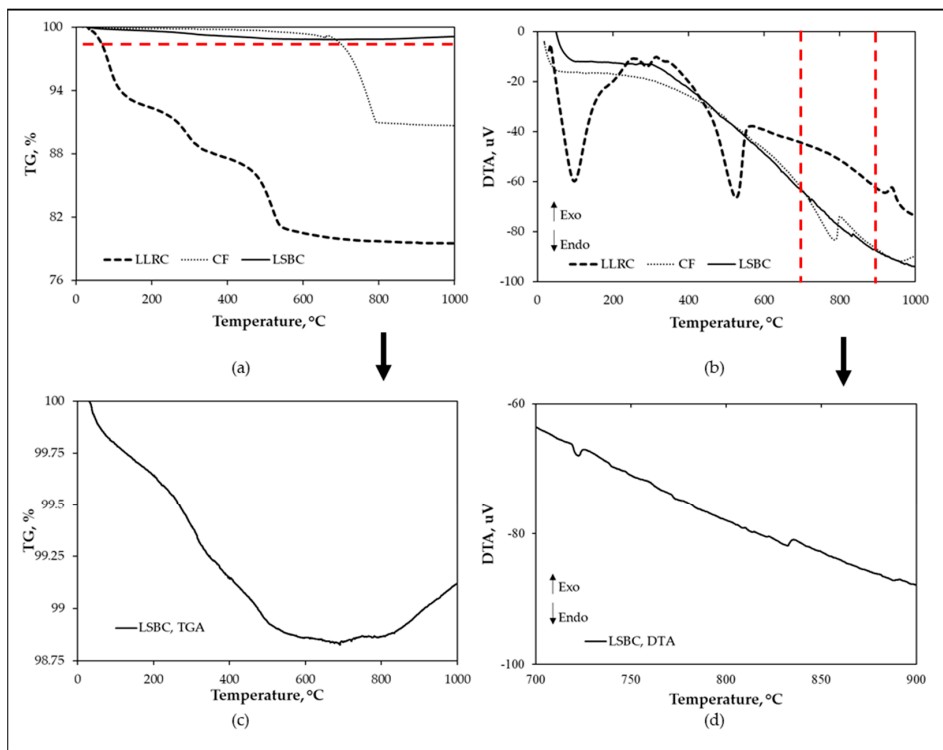

**Figure 4.** TGA-DTA of raw materials: (**a**) TGA; (**b**) DTA; (**c**) magnified portion of TG curve of LSBC; and (**d**) magnified portion of DTA curve of LSBC at 700–900 °C.

The XRD of LSBC showed a predominance of feldspar, particularly andesine and anorthite. Thus, proximate analysis of the available chemical analysis of LSBC was carried out using feldspar convention [12]. The results showed high amounts of feldspar and free silica with approximate amounts of 42.52 wt% and 35.32 wt%, respectively, while the clay content is approximately 3.69 wt%. Other oxides such as $Fe_2O_3$, NiO, $Cr_2O_3$, MnO and $TiO_2$ amounted to approximately 18.46%. The decomposition of the hydroxyls could be attributed to the minor phases, specifically smectites such as montmorillonite [65],

nontronite [66], illite [67] and lizardite [68]. The decomposition of the hydroxyls of the various smectites caused the loss of mass which resulted in a drop of the TGA curve at temperatures of 470 °C for nontronite [66], 550 °C for montmorillonite [65], 580 °C for lizardite [68] and 724 °C for illite [67]. The low drop of the mass loss of LSBC according to the feldspar convention is approximately 0.52 wt%, which contributed to the 1.15 wt % mass loss before reaching 700 °C. A continuous loss of mass for the smectites could be observed beyond 700 °C [65–68]. A slight, gradual weight gain was observed as the temperature was further increased from 700 °C to 1000 °C, which could be due to the oxidation and allotropic transformation of the iron oxide content of LSBC, such as nontronite, illite, and augite. The simultaneous loss of weight by the smectites was outweighed by the weight gain of the aforementioned oxidation, thereby showing an apparent gain of mass in the TGA-DTA of LSBC. As seen in the magnified TGA of LSBC (Figure 4c), a mass loss of approximately 1.15% is observed around 700 °C. This is followed by a gradual mass gain of approximately 0.28% until it reached a temperature of 1000 °C.

The thermogravimetric behavior of the formulated powders was compared to their raw material counterparts. Figure 5a,b shows the DTA curves and mass losses obtained from the thermogravimetric analysis of the formulated powders (LF, LFBQ, LFBH, and LB). All formulated powders displayed a similar behavior on their DTA and TG curves except for sample LB, which had the full replacement of commercial feldspar by the LSBC. It can be seen that the total mass loss of LF, LFBH, LFBQ and LB are 14.48%, 12.83%, 11.78% and 7.26%, respectively. The mass loss decreased as LSBC was increased, which concurs with the TGA result of LSBC. In the DTA curves (Figure 5b), three (3) endothermic peaks and two (2) exothermic peaks were identified. The first region shows an endothermic reaction occurred at 73.29 °C, 76.32 °C, 75.83 °C and 75.35 °C for LF, LFBQ, LFBH and LB, respectively, and is related to the release of free and adsorbed water [12,43,56,57]. The second region presents an exothermic peak observed at 315.91 °C for LF, 313.97 °C for LFBQ, 317.18 °C for LFBH, and 309.48 °C for LB, which could be attributed to the burning off of carbonaceous matter [57] from LLRC. The third region is an endothermic peak observed at 512.62 °C for LF, 504.75 °C for LFBQ, 509.85 °C for LFBH and 504.97 °C for LB, associated with the loss of the hydroxyl group from the red clay (LLRC) [43,56,57]. The fourth region are endothermic reactions occurring at 775.31 °C for LF, 765.85 °C for LFBQ and 753.00 °C for LFBH, which can be attributed to the calcite decomposition [61,62] accompanied by significant loss of mass as emphasized in Figure 5c. No distinct phase transformation was observed in the DTA curve of LB at temperatures above 750 °C, as shown in Figure 5d, which is consistent with the DTA of LSBC, as shown in Figure 4. The decomposition of montmorillonite and formation of defect such as aluminum-silicon spinel ($Si_3Al_4O_{12}$) structures (structural reorganization), and some amorphous phases [57–60] are attributed in the last region on the DTA curve, which is an exothermic peak at 923.41 °C, 930.25 °C, 936.94 °C, and 936.03 °C for LF, LFBQ, LFBH and LB, respectively. At these high temperature levels, the Fe ions in this region caused the clay to dissolve and disperse [69,70]. Since LSBC has a higher iron content than CF, a distinct difference of the porosity and water absorption was observed with LB, which is shown in Figure 6.

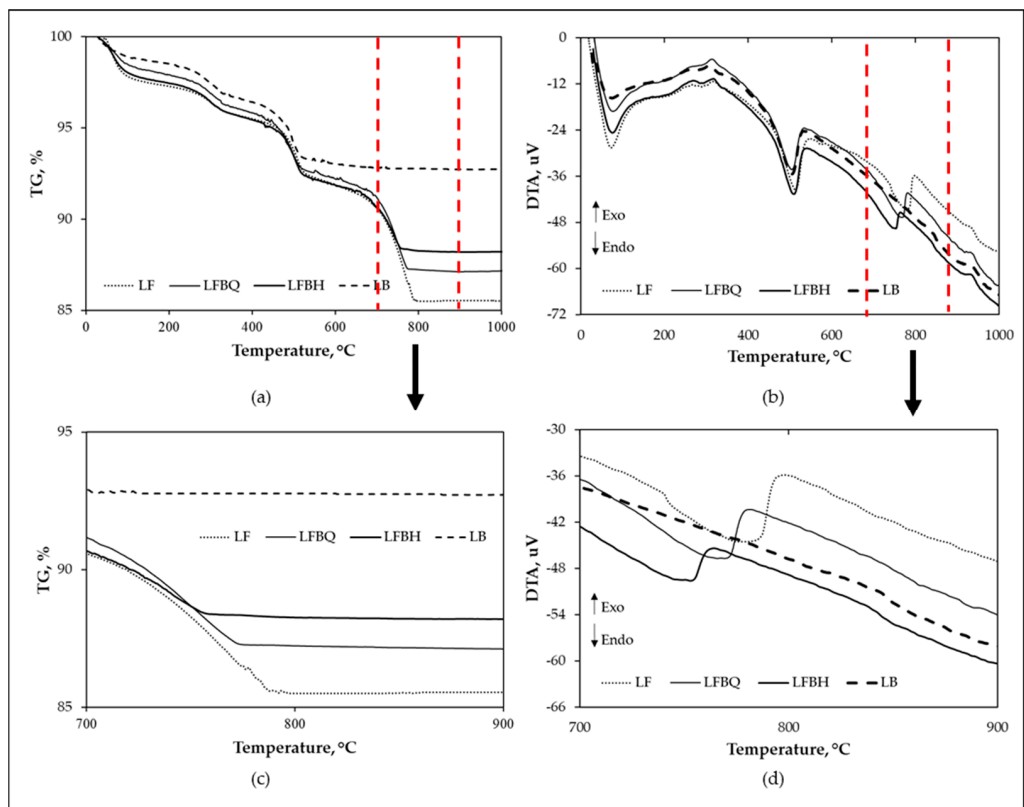

**Figure 5.** Analysis of different formulations: (**a**) TG; (**b**) DTA; (**c**) portion of TG curve (**a**) showing the different behaviors of the formulations at 700–900 °C; and (**d**) portion of DTA curve (**b**) showing the different behaviors of the formulations at 700–900 °C.

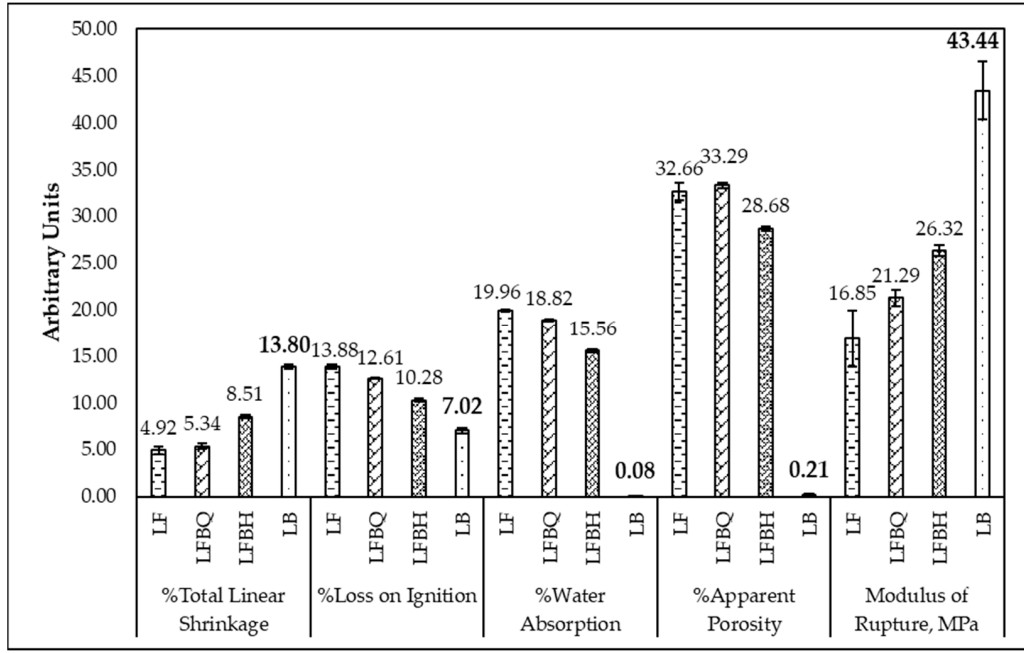

**Figure 6.** Physical and mechanical properties of tiles fired at 1200 °C.

### 3.3. Physical and Mechanical Properties

Test bars produced from the different formulations were further characterized in terms of their physical and mechanical properties. The average results on total linear shrinkage, loss on ignition, water absorption, apparent porosity, and modulus of rupture are shown

in Figure 6. The average total linear shrinkage increases as the LSBC content is increased, which could be attributed to its low silica content, as shown in Table 2. The loss on ignition decreases as the LSBC content is increased, which means the high loss on ignition of the test bars is primarily contributed by LLRC and CF, and is thus consistent with the TGA behavior of LLRC and CF shown in Figure 4a. In addition to reducing the loss on ignition, there is also further reduction of the water absorption and apparent porosity upon increasing the amount of LSBC. Thus, lowering the water absorption and apparent porosity is evident when LSBC partially replaced commercial feldspar, starting with LFBQ exhibiting a higher MOR than LF. Such that LB bars with full replacement of feldspar with LSBC obtained the highest MOR of 43.44 MPa. LB bars were highly vitreous, exhibiting a low average water absorption of less than 0.08%. The higher iron content of LSBC than CF could have resulted in more dissolution of clay [69,70] in the bar matrix, which further increased the formation of a large amount of liquid phase which penetrated the structure's open pores and resulted in densification by liquid phase sintering [44,71]. The densification of these pores will increase the solid matter in the matrix of the bars in resisting any applied load to fracture the tiles, consequently increasing its strength against fracture or its modulus of rupture. Moreover, bars from this study have lower water absorption and higher MOR than NMW tiles [45] even if the latter has higher flux/silica ratio since those were fired at a lower temperature.

The foregoing results of black cinder in partially and fully replacing feldspar in red stoneware pose promising results in achieving properties such as low porosity, low water absorption and high MOR. Tiles made from LB which are fired at 1200 °C are shown in Figure 7. The fired color is in the range of (7.5 YR, 4/6) strong brown and (5 YR, 5/6) yellowish red [72], and has a shiny and opaque luster. The other formulations did not exhibit shiny luster, which gives LB tiles a distinct aesthetic appearance among the other tiles.

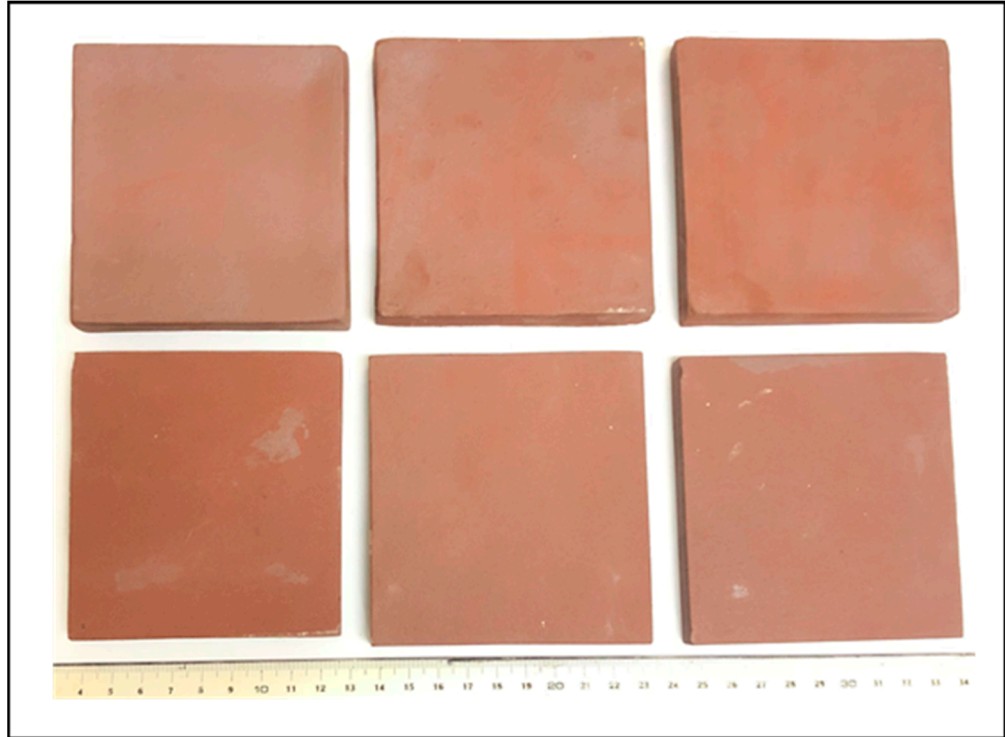

**Figure 7.** LB Tiles sintered at 1200 °C.

### 3.4. Potential Engineering Application

A summary of the water absorption and MOR requirements of the various standards are shown in Table 4. Based on the results of LF, LFBQ and LFBH, as seen in Table 5, all

passed the requirements for wall tile application as they passed the minimum requirement for ISO standard 13006 AIII (>10% water absorption and >8 MPa MOR), PNS (>15 MPa MOR), and ICCTAS (>10 and <20% water absorption and >15 MPa MOR). Moreover, LFBH passed the MOR requirement of ICCTAS for floor tile application, which is greater than 22 MPa. All formulated tiles have potential for roof application since they passed the ABNT minimum requirements on water absorption (<20%) and breaking strength (100 kgf). LB samples which have full replacement of CF with LSBC, passed the water absorption and MOR requirements of the four standards which means that these samples are suitable for wider applications such as wall, floor, vitrified, industrial and roof applications.

**Table 4.** Various Standard Requirements for Ceramic Tiles on Water Absorption and MOR.

| Application | Water Absorption, % | Modulus of Rupture, MPa | Source |
|---|---|---|---|
| Floor Tile (Porcelain Stoneware) | <0.5 | >35 ± 2 | ISO standard 13006 AIa [37] |
| Wall Tile | >10 | >8 | ISO standard 13006 AIII [37] |
| Floor Tile<br>Wall Tile | <0.5<br>NA | >35<br>>15 | PNS 154:2005 [38] |
| Floor Tile<br>Wall Tile<br>Vitrified Tile<br>Industrial Tile | >3 and ≤6<br>>10 and <20<br>≤0.5<br>≤0.5 | ≥22<br>≥15<br>≥32<br>≥32 | ICCTAS ESTD 1990 [39] |
| Roof Tile | <20 | 100 kgf (breaking strength) | ABNT NBR 15310:2005 [35,36] |

**Table 5.** Quality compliance summary.

| Formulation for the Low-Porosity Red Stoneware | This Study, 2022 | | ISO Standard 13006 [37] | | PNS 154:2005 [38] | | ICCTAS ESTD 1990 [39] | | | | ABNT NBR 15310:2005 [35,36] |
|---|---|---|---|---|---|---|---|---|---|---|---|
| | **WA** | **MOR** | **Floor** | **Wall** | **Floor** | **Wall** | **Floor** | **Wall** | **Vitrified** | **Industrial** | **Roof** |
| LF | 19.96 ± 0.11 | 16.85 ± 3.05 | X | ✓ | X | ✓ | X | ✓ | X | X | ✓ |
| LFBQ | 18.82 ± 0.12 | 21.29 ± 0.90 | X | ✓ | X | ✓ | X | ✓ | X | X | ✓ |
| LFBH | 15.56 ± 0.21 | 26.32 ± 0.60 | X | ✓ | X | ✓ | X | ✓ | X | X | ✓ |
| LB | 0.08 ± 0.03 | 43.44 ± 3.06 | ✓ | ✓ | ✓ | ✓ | ✓ | ✓ | ✓ | ✓ | ✓ |

Note: ✓: passed the quality requirement X: failed the quality requirement.

## 4. Conclusions and Future Works

The findings presented and discussed in this paper demonstrate promising results of LSBC as partial and full replacement of feldspar in addition to locally sourced red clay (LLRC) for the production of low-porosity red stoneware. Utilization of LSBC as a raw material in ceramic applications will ensure higher economic value than its present use as a filling material, as well as contribute to lowering the production cost of the traditional ceramic mixture as it will lessen the importation of feldspar into the country. Based on the results of the study, the following conclusions are drawn:

- The chemical composition of Lanao Salvador black cinder (LSBC) shows comparable content of fluxes as with commercial feldspar.
- The thermal behavior of the formulated bodies exhibited similar behavior in their DTA and TG curves to their raw material counterparts except for the LB sample, which has the full replacement of commercial feldspar with LSBC.
- Full replacement of feldspar with LSBC obtained the highest MOR of 43.44 MPa. Increasing the LSBC content during its partial substitution for feldspar resulted in an

increase in total linear shrinkage and a decrease in other physical properties such as loss on ignition, water absorption and apparent porosity.

- Full replacement of feldspar with black cinder (LB) is suitable for broader application such as wall, floor, vitrified, industrial and roof tiles. Partial replacement of feldspar with black cinder (LF, LFBQ and LFBH) is feasible for wall and roof tile applications.

The following are the recommendations for future works:

- Conduct of other tests such as crazing resistance, thermal shock, abrasion resistance, chemical resistance, stain resistance, scratch hardness and frost resistance on ceramic tiles for floor, wall, vitrified and industrial applications.
- Conduct of transverse strength test, frost resistance and pyroplastic deformation for roofing tile application.

**Author Contributions:** Methodology, F.J.A.E., L.R.L., B.L.B., A.M.R.S., I.C.B.-A., R.V.R.V., J.P.C., E.U.A.J., R.V.M.D., S.K.D.D. and C.J.C.S.; formal analysis, F.J.A.E., L.R.L., B.L.B., A.M.R.S., I.C.B.-A. and R.V.R.V.; software, F.J.A.E., I.C.B.-A. and R.V.R.V.; validation, F.J.A.E., L.R.L., B.L.B., R.V.R.V. and I.C.B.-A.; investigation, F.J.A.E., L.R.L., B.L.B., A.M.R.S., J.P.C., E.U.A.J., R.V.M.D., S.K.D.D., C.J.C.S., R.V.R.V. and I.C.B.-A.; resources, F.J.A.E., L.R.L., B.L.B., A.M.R.S., J.P.C., E.U.A.J., R.V.M.D., S.K.D.D., C.J.C.S., R.V.R.V. and I.C.B.-A.; data curation, F.J.A.E., L.R.L., B.L.B., A.M.R.S., R.V.R.V. and I.C.B.-A.; writing—original draft preparation, F.J.A.E., L.R.L., B.L.B., A.M.R.S., R.V.R.V. and I.C.B.-A.; writing—review and editing, F.J.A.E., L.R.L., B.L.B., A.M.R.S., R.V.R.V. and I.C.B.-A.; visualization, F.J.A.E., L.R.L., B.L.B., A.M.R.S., R.V.R.V. and I.C.B.-A.; supervision, F.J.A.E., L.R.L., R.V.R.V. and I.C.B.-A.; project administration, I.C.B.-A., R.V.R.V. and L.R.L.; funding acquisition, I.C.B.-A. and R.V.R.V.; conceptualization, I.C.B.-A. and R.V.R.V. All authors have read and agreed to the published version of the manuscript.

**Funding:** This research was funded by the Department of Science and Technology, Philippine Council for Industry, Energy, and Emerging Technology Research and Development (DOST-PCIEERD), under grant number 7374.

**Data Availability Statement:** Not applicable.

**Acknowledgments:** The authors acknowledge the following organizations: Ceramic Engineering and Department of Materials and Resources Engineering and Technology of Mindanao State University—Iligan Institute of Technology and the Department of Science and Technology Philippine Council for Industry, Energy, and Emerging Technology Research and Development (DOST-PCIEERD). Acknowledgement is also extended to Engr. John Louie L. Tefora for his administrative and laboratory assistance.

**Conflicts of Interest:** The authors declare no conflict of interest. The funders had no role in the design of the study; in the collection, analyses, or interpretation of data; in the writing of the manuscript; or in the decision to publish the results.

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
