# Peer review of "Effect of Replacing Feldspar by Philippine Black Cinder on the Development of Low-Porosity Red Stoneware"

_minerals, doi:10.3390/min13040505_

Round 1

Reviewer 1 Report

Dear Authors, the manuscript revised   reports an interesting study regards the feasibility to substitute the commercial feldspar by Black Cinder from Philipines in line with the Circular Economy and the valorisation of the residues .

Regarding the mineralogical results invite the authors to check the presence of albite in commercial feldspar if the Na2O is not present in the chemical analysis and suggest the authors include a SEM photo for the feldspar.

I attach a file with the revised manuscript with my corrections and questions,

Reviewer 2 Report

There is not good correlation between chemical and XRD data. It must be revised. The relative abundances of minerals in raw materials should be determined.

Line 15:”is” instead of “are”

Line 109: Change “The mineralogical phases were carried out by X-ray diffraction” by “The mineralogical phase characterization was carried out by X-ray diffraction”

Line 94: Add “( LSBC)” after “ Lanao Salvador black cinder”

Line 128: Add “commercial feldspar “ before CF in  “LSBC and CF”:

Line187: “CF contains minerals such as albite, calcite and quartz”. This is coherent with XRD data but does not match with chemical data presented in Table 2. Albite (NaAlSi3O8), quartz (SiO2) and calcite (CaCO3) do not contain Mg. Chemical data does not show Na in CF.

Figure 1: LLRC pattern is poorly defined. Peaks are not clear.

Figure 3: Montmorillonite ((Na, Ca)0.3 (Al, Mg)2 Si4 O10 (OH)2 • n H2O), Illite

(K,H3O)(Al,Mg,Fe)2(Si,Al)4O10[(OH)2,(H2O)], Nontronite

Ca.5(Si7Al.8Fe.2)(Fe3.5Al.4Mg.1)O20(OH)4 and Lizardite Mg3Si2O5(OH)4, should lose hydroxyls if present and detectable in LSBC. The mass loss of LSBC is not coherent with de presence of these minerals.

Round 2

Reviewer 1 Report

Dear Authors,

the new version of the manuscript is improved following my and the other reviewer's suggestions. Now, the level has been enhanced; as required, more information and references have been inserted in the text. 

In my opinion, using an instrument that does not detect sodium oxide for the characterization of ceramic raw materials should be detailed in the experimental paragraph.

Regards
